

# Monthly-Scale Extended Predictions Using the Atmospheric Model Coupled with a Slab-Ocean

Zhenming Wang[1,2], Shaoqing Zhang[*1,2,3], Yishuai Jin[1,2], Yinglai Jia[1], Yangyang Yu[1,2], Yang Gao[3,4], Xiaolin Yu[1,2,3], Mingkui Li[1,2,3], Xiaopei Lin[1,2,3], Lixin Wu[1,2,3],

[1] Key Laboratory of Physical Oceanography, Ministry of Education, Institute for Advanced Ocean Study, Frontiers Science Center for Deep Ocean Multispheres and Earth System (DOMES), Ocean University of China, Qingdao, 266100, China
[2] College of Oceanic and Atmospheric Sciences, Ocean University of China, Qingdao, 266100, China
[3] Pilot National Laboratory for Marine Science and Technology, Qingdao, 266100, China
[4] Key Laboratory of Marine Environment and Ecology, and Frontiers Science Center for Deep Ocean Multispheres and Earth
System (FDOMES), Ministry of Education, Ocean University of China, Qingdao 266100, China

*Correspondence to:* Shaoqing Zhang *(szhang@ouc.edu.cn)*

**Abstract.** Given the good persistence of sea surface temperature (SST) due to the slow-varying nature of the ocean, an atmospheric model coupled with a Slab Ocean Model (SOM) instead of a 3-D dynamical ocean model is designed as an efficient approach for extended-range predictions. The prediction experiments from July to December 2020 are performed
based on the Weather Research and Forecasting (WRF) model coupled to the SOM (WRF-SOM) with the initial and boundary conditions same as the WRF coupled to the Regional Ocean Model System (WRF-ROMS). The WRF-SOM is verified to have better performance of SSTs in the extended-range predictions than WRF-ROMS since it avoids the complicated model biases from the ocean dynamics and seabed topography when extended-range predictions are made using a 3-D dynamical ocean model. The improvement of SSTs can lead to the remarkable impact on the response of the
atmosphere from the surface to the upper layer. Taking typhoon as an example of extreme events, the WRF-SOM can obtain comparable intensity predictions and slightly improved track predictions due to the improved SSTs in the initialized WRF-SOM system. Overall, the WRF-SOM can ensure the stability of extended-range prediction and reduce the demand for computing resources by roughly 50%.





## 1 Introduction

Extended-range predictions fill the gap between weather and climate predictions. Recent research has demonstrated kinds of
sources of predictability for the extended period such as the Madden-Julian Oscillation (MJO), the evolution of El Niño-
Southern Oscillation, soil moisture, snow cover, sea ice, stratosphere-troposphere interactions, ocean conditions, and
tropical-extratropical teleconnections (Wheeler and Hendon, 2004; Vitart and Robertson, 2018). As a growing demand from
the applications community and progress in identifying and simulating key sources of extended period (Vitart, 2014; White
et al., 2017), it is worthwhile improving forecast skills for monthly-scale extended period predictions to realize the social
security, disaster early warning, agricultural management, and water resource management (David, 2010).

In the extended period prediction, sea surface temperature (SST) is the most important information provided by the oceanic
model to the atmospheric model in the air-sea interaction. For instance, tropical SST plays an important role in controlling
the weather/climate worldwide by various teleconnection effects (David, 2010). Dian et al. (2013) demonstrated the
importance of air-sea interaction to the atmospheric mesoscale processes by comparing the response of the precipitation to
the SST between the coupled and uncoupled models. Furthermore, Stan (2018) emphasized that the SST anomaly can
directly lead to the change in the convection intensity.

The ocean-atmosphere coupling has an important impact on the extended-range prediction skills (Vitart and Molteni, 2010).
Rashid et al. (2019) adopted the Bureau of Meteorology unified atmospheric model (BAM) coupled with the Australia
Community Ocean Model (ACOM) to predict the MJO and proposed that actual MJO prediction skills may be further
improved through continued development of the dynamical prediction system. The coupled ocean-atmosphere models are
mainly used for numerical simulation and prediction in the extended period (Saravanan and Chang, 2019).

However, there are some inherent defects for the specific problems in extended-range prediction using atmosphere-ocean
coupled models. For instance, the 3-D dynamical ocean model inevitably introduces unnecessary biases from the seabed
topography, which can transport from bottom to surface during prediction (Wu et al., 1997). The 3-D dynamical ocean
model coupled to the atmospheric model can have cold drift during the extended-range prediction period due to the
overestimation of latent heat in the coupled model (Ren and Qian, 2010). In addition, the sensitivity of ocean
thermodynamics to the ocean dynamics leads to the enhancement of mixing in the upper ocean and indirectly reduces SST
(Hu et al., 2017). European Centre for Medium-Range Weather Forecasts (ECMWF) summarized and evaluated the results
during the extended period prediction, and proposed that the improvement of extended-range prediction should be
accompanied by the significant reduction of SST biases in a coupled model (Palmer et al., 1990).

Considering that SST is the most important factor provided by the ocean model and 3-D dynamical ocean models have a
deficiency in the SST prediction during the extended period, one possible way to improve this period prediction is that, we
only focus on the SST as the bottom boundary of atmospheric model for the extended-range prediction research. The SST
has good persistence in the extended period and only the thermal effect needs to be considered (the time scale of ocean
circulation is relatively long). According to that, the Slab Ocean Model (SOM) can be utilized as the ocean model for





extended-range prediction such that biases of SST are easier to manage (Zuidema, 2016). More importantly, the SOM can greatly reduce the computing expense and obtain the forecast results more quickly, which can provide a more economical and efficient method for further study.

In this paper, we develop a new approach using atmospheric model coupled with a Slab Ocean Model (WRF-SOM) to do monthly-scale extended-range predictions. For comparison of the prediction results of WRF-SOM, we also carry out the forecast experiments using WRF coupled to the Regional Ocean Model System (ROMS) based on the regional coupled prediction system for the Asia-Pacific (AP-RCP) developed by Li et al. (2020). Firstly, by comparing the performances of SST predictions in the WRF-SOM and WRF-ROMS, we show the rationality of WRF-SOM in the extended-range predictions. WRF-SOM can avoid the influence of cold deviation at the subsurface in WRF-ROMS on SST in extended period. Secondly, we discuss the response of atmosphere (e.g., the air temperature, and geopotential height) on SSTs to identify the improvement of WRF-SOM compared with WRF-ROMS in the cold deviation area. Finally, taking typhoons as the representation of the extreme weather events, we track the differences of typhoon paths and maximum wind speed (MWS) between WRF-ROMS and WRF-SOM and suggest that the performances of typhoon predictions are basically consistent in the two models during the extended period.

The rest of the paper is organized as follows. Section 2 details the source of SST biases, the brief introduction of WRF-SOM and WRF-ROMS, the experiment implementation, and the data sources. Section 3 evaluates the feasibility of SST predictions in WRF-SOM, compares the response of the atmosphere to SSTs in WRF-SOM and WRF-ROMS, and verifies the rationality of WRF-SOM in typhoon predictions. Finally, the summary and discussion are given in Section 4.

## 2 Methodology

### 2.1 Brief introduction of WRF-ROMS coupled model

In this study, we use the high-resolution WRF-ROMS coupled system for comparison (Li et al., 2020). The system covers the area of the Asia-Pacific, which consists of 27 km WRF, 9 km ROMS, and observational information through dynamically downscaling coupled assimilation. The vertical layers of WRF and ROMS are 28 and 33 respectively. The time step for both WRF and ROMS is 60 s, the coupled interval time between ocean and atmosphere is 600 s, and the forecast lead time is 34 days for each case. The system is initialized from the Climate Forecast System Version2 reanalysis (CFSv2) (Saha et al., 2014), on January first, 2016, and spun up for two years. The system is verified for the precipitation forecast skills, which is the highest among CFSv2, National Centers for Environmental Prediction-Global Ensemble Forecast system (NCEP-GEFS) (https://www.nco.ncep.noaa.gov/pmb/products/gens/), and European Centre for Medium-Range Weather Forecasts-Ensemble Prediction System (ECMWF-EPS) (https://www.ecmwf.int/en/forecasts/datasets). The operational system has realized the extended-range prediction of atmospheric and oceanic environments and serves as an effective research platform to study the influence of model resolution on typical mesoscale atmospheric and oceanic phenomena in the



Asia-Pacific area. The high-resolution prediction system enhances the capability of atmosphere-ocean coupled models to describe many local details, which is a necessary step to discuss the predictability in the extended period.

## 2.2 Slab-Ocean scheme in a coupled model

In order to describe the response of the upper ocean to the surface wind, a simple model (SOM) is given (Raymond et al., 1973). Compared with the 3-D dynamical ocean model, the ocean mixed layer temperature is the only prognostic state variable for the SOM to represent the SST. Jia et al. (2019) adopted the SOM to study the ocean mesoscale variability. The related prognostic equation is the first law of thermodynamics for the ocean mixed layer given by Eq.(1):

$$\rho C_p h_{mix} \frac{\partial T_{mix}}{\partial t} = Q_{atm} - Q_{ocn} , \tag{1}$$

where $\rho$ is the ocean water density, $C_p$ is the specific heat capacity of the ocean water, $h_{mix}$ is the depth of the mixed layer, $T_{mix}$ is the mixed layer temperature, $Q_{atm}$ is the net surface heat flux from the atmosphere to the mixed layer, and $Q_{ocn}$ is the net heat transfer from mixed-layer column to the subsurface. Eq.(2) shows the heat budget of the sea surface from the atmosphere:

$$Q_{atm} = Q_{sol} - Q_l - Q_{sen} - Q_{latent} , \tag{2}$$

where $Q_{sol}$ is the net radiative heating of the ocean mixed layer by solar radiation, $Q_l$ is the net longwave radiative cooling of the ocean mixed layer, $Q_{sen}$ is the net sensible heat flux from the ocean to the atmosphere, $Q_{latent}$ is the net latent heat flux from the ocean to the atmosphere. $Q_{atm}$ and $Q_{ocn}$ are calculated synchronously with the prediction time in the model. Eq.(3) shows the effect of Coriolis force and wind stress in the mixed layer:

$$\begin{array}{l} \dfrac{\partial h_u}{\partial t} = f \times h_v + \tau_x \\[2mm] \dfrac{\partial h_v}{\partial t} = -f \times h_u + \tau_y \end{array} , \tag{3}$$


where $h_u$ ($h_v$) is the $\tau_x$-driven ($\tau_y$-driven) momentum in the ocean mixed layer, $f$ is Coriolis force, and $\tau_x$ and $\tau_y$ are respectively the zonal and meridional components of wind stress at the surface. Eq.(3) is calculated by time-centering difference and Eq.(4) shows the variations of ocean mixed layer depth, which is affected by the wind stress and heat flux:


$$h_{mix} = \frac{Q_{atm} - Q_{ocn}}{\Gamma} + \sqrt{\frac{(Q_{atm} - Q_{ocn})^2}{\Gamma^2} + \frac{2 \times (h_{u2}^2 + h_{v2}^2)}{\Gamma g \alpha}} , \tag{4}$$

where $h_{mix}$ is the mixed layer depth, $g$ is the gravitational acceleration, $\Gamma$ is the lapse rate of the water temperature, and $\alpha$ is the thermal expansion coefficients.

Such basic driving processes of WRF-SOM and the relationship between the variables can be illustrated in **Fig. 1**. The SOM is driven by the surface wind, sea surface heat flux, and heat conduction between the mixed layer and subsurface. The mixed



layer depth is determined by the surface wind stress ($\tau_x$ and $\tau_y$) and the heat budget ($Q_{atm}$ and $Q_{ocn}$) in the mixed layer. Both the enhancement of surface wind stress and heat flux to the mixed layer will lead to the deepening of the mixed layer depth. When the ocean surface is heated, there will be a temperature gradient from from sea surface to the areas beneath it. With the wind stirring the upper layer, an almost uniform layer is formed, and there is a density gradient below the mixed layer. In the upper mixed layer, the temperature is independent of depth. We assume that once the initial delamination is destroyed in this layer, it will mix to a completely uniform state. It means that the ocean temperature is well-mixed and the SST is considered the same as $T_{mix}$.

## 2.3 Implementation, data source (including model setting and initial condition sources), and data processing method

In this study, the WRF version WRF3.7.1 and the ROMS version ROMS3.8 is applied (Skamarock et al., 2008; Shchepetkin and Mcwilliams, 2005). The boundary condition of the forecast is interpolated from the CFSv2 forecast data set. The WRF-ROMS is initialized from the CFSv2 reanalysis at 00 UTC on 1st January 2016, spun up for two years with the CFSv2 background boundary conditions, and applies the weakly coupled data assimilation approach (WCDA). The WCDA begins after a 2-year spin for the coupled model. In addition, the atmospheric and oceanic components are restrained by the cycling real-time operational data, which provides the initial conditions for the usual forecast. The simulation region covers the Asia-northwest Pacific and North Indian Ocean (74° E-180° E, 18° S-60° N). The forecasts are made every day from July 19th to December 31st 2020. Each case generates a 34-day forecast for the atmosphere and ocean environment.

The WRF-SOM is completely consistent with WRF-ROMS in atmospheric model settings and the grid of SOM is consistent with the atmospheric model. The forecast cases made by WRF-SOM are same to the WRF-ROMS expect for roughly 20 cases/days are missed, which is caused by hardware damage and untimely release of boundary information. Li and Ding (2011) proposed that the linear relationship between the predictability limit and the logarithm of initial error holds only in the case of relatively small initial errors. If the initial errors are large, the growth of mean error would directly enter into the nonlinear phase. Therefore, for each example, we keep the initial and boundary conditions of the WRF-SOM in the atmosphere and ocean the same as those in the WRF-ROMS and assure that the forecast lead time of each example is over one month.

The Hybrid Coordinate Oceanic Circulation Model (HYCOM) reanalysis (https://www.hycom.org) used in this study is provided by Naval Research Laboratory (Cummings and Smedstad, 2013). Considering the HYCOM reanalysis being a mature and widely recognized forecast system, the global reanalysis data can be a good choice to verify the model prediction performances. The typhoon observations are from the National Meteorological Center (NMC) of China (http://typhoon.nmc.cn/web.html). The validation data of the atmospheric component is from CFSv2 (https://rda.ucar.edu/datasets/ds094.1/). All the simulation experiments use the computing nodes configured with 24 central processing unit (CPU) cores, 2.6 GHz dominant frequency, and 256 GB of global DDR4 memory.





The predictability of SST in WRF-ROMS and WRF-SOM is evaluated by the root mean square error (RMSE) the anomaly correlation coefficient (ACC), which is written as follows:

$$RMSE_j = \sqrt{\frac{1}{M}\sum_{i=1}^{M}(x_{i,j} - f_{i,j})} \quad , \tag{5}$$

$$ACC_j = \frac{\sum_{i=1}^{M}(x_{i,j} - \overline{x_j}) \times (f_{i,j} - \overline{f_j})}{\sqrt{\sum_{i=1}^{M}(x_{i,j} - \overline{x_j})^2 \times \sum_{i=1}^{M}(f_{i,j} - \overline{f_j})^2}} \quad , \tag{6}$$

where $x_{i,j}$ is the forecast value, $f_{i,j}$ is the truth value (reanalysis data), $\overline{x_j}$ is the spatial average of the forecast value, $\overline{f_j}$ is the spatial average of the truth value, and $i = 1,2,3...M$ and $j = 1,2,3....N$ represent grid points and time series respectively.

## 3 Comparing the forecast results of WRF-SOM with WRF-ROMS

### 3.1 Predictability of sea surface temperature and bias

In the 3-D dynamical ocean model, the SST prediction is affected by topography accuracy and ocean dynamics. Wu et al. (1997) suggested that due to the existence of seabed topography with finite amplitude, the wave models in a linear system are no longer independent of each other, resulting in coupling between models. This coupling effect between models acts on the circulation field in different ways, making the simple linear superposition of models no longer truly reflect the oceanic circulation field structure. The effects of seabed topography on the baroclinic model should be stronger. Therefore, the seabed topography can indirectly affect the SST through the circulation field. As for ocean dynamics, the inaccuracy of processes (advection, vertical mixing, and vertical diffusion) and atmosphere-land model jointly cause SST deviations (Hu et al., 2017). The model resolution is another way affecting the SST prediction, which is verified that the biases can be slightly eliminated in the Kuroshio extension area with the increase in model resolution (Li et al., 2020). Therefore, based on the good persistence of SST, we can simplify the SST evolution process to avoid biases from the ocean dynamics and seabed topography (Zuidema et al., 2016).

The prediction skills of SST in the WRF-SOM and WRF-ROMS have been assessed by calculating the RMSE averaged of 142 forecast cases from July to December. **Figure 2a** shows the RMSE of SSTs in the WRF-SOM is generally lower than that in the WRF-ROMS and both forecast errors increase with the lead time. The maximum different value of the RMSEs variation between the WRF-SOM and WRF-ROMS occurs in 20-25 days. The averaged values of the SST errors in both models are within 1.4℃ during the forecast period and the RMSEs of 75% forecast cases in WRF-SOM are better than those in WRF-ROMS, as shown in **Fig. 2b**. Moreover, the bias in the WRF-SOM grows more slowly than that in the WRF-ROMS. Only at the start of the forecasts, the errors of SST in WRF-ROMS are lower than that in WRF-SOM. It is because the errors in the 3-D dynamical ocean model have not spread from subsurface to the surface and initial condition still plays a major





role (Lekshmi et al., 2022). To explore the spatial distribution of skills with different forecast periods in WRF-SOM, we use

the ACC of SST anomaly to characterize the temporal and spatial predictability of SST in the WRF-SOM and WRF-ROMS (Wu et al., 2016). **Figure 2c** shows that their overall ACC can reach more than 0.75 during the 34-day forecasts and the performance of the WRF-SOM in the whole domain is higher than that of WRF-ROMS. Meanwhile, the ACC in 74% forecast cases in WRF-SOM is better than those in WRF-ROMS, as shown in **Fig. 2b**. Finally, **Figure 3a** and **3b** show the forecast SSTs in WRF-ROMS have an obvious cold deviation in the area around the Kuril Islands and the Sea of Okhotsk

(35° N-58° N, 140° E-160° E).

In order to explore the spatial distribution of SST prediction skills in the two models, especially in the cold deviation area, we calculate ACC of SSTs at each grid point. Through the spatial distribution of ACC displayed in **Fig. 4** in different forecast periods, it is found that the predictability of WRF-SOM and WRF-ROMS decreases with time in the whole area. Since the initial state of the ocean can be maintained for a period of time in the simulation, the main patterns of the ACC are

consistent in the two models, and the value increases with the latitude significantly. The higher skills of WRF-SOM are mainly concentrated in the area north of 15°N compared with WRF-ROMS. Over 60% grid points in the simulation area have higher ACC values of SST in WRF-SOM, and the proportion rises slightly with the prediction time (green dots in the right column of **Fig. 4**). Focused on the cold deviation area in the green rectangle, the proportion reaches more than 80% (red dots in the right column of **Fig. 4**). In summary, the performance of SSTs in the WRF-SOM is more reasonable than the

WRF-ROMS in terms of temporal variation and spatial distribution of predictability.

In order to explore the causes of cold deviation area in WRF-ROMS, the variations of averaged error of SST and the water temperature at the subsurface are shown. **Figure 5a-5c** identifies the comparison of averaged mixed layer depth during the prediction period. The mixed layer depth in WRF-ROMS is calculated by the depth at which the difference from SST is 0.2 ℃. The mixed layer depth in WRF-ROMS is significantly greater than that in WRF-SOM and the reanalysis data from

ECMWF. Moreover, the cold deviation of WRF-ROMS at the subsurface continues to conduct upward with the forecast time, and finally the predicted value of SST is low in this area as shown in **Fig. 5d** and **5e**. The abnormal cold deviation at the subsurface is caused by the imprecise description of the ocean processes and the insufficient resolution of the 3-D dynamical ocean model. In addition, the data assimilation can accelerate the heat loss and intensify the cooling in this area. By eliminating the influence of initial conditions and the oceanic heat transport, the WRF-SOM can obtain better SST qualities

with the avoidance of biases from the model dynamics and inaccurate seabed topography.

**3.2 Impact on Extended-range Predictions**

The SST differences between WRF-ROMS and WRF-SOM spread rapidly in all prediction cases and have obvious thermodynamic feedback to the atmosphere. For instance, the region with a large deviation of SST is expected to have a great impact on the atmospheric process (Hao et al., 2016). The first mode of SST and air temperature at 850 hPa is verified

to be positively correlated in most of the East China Sea (Zeng et al., 2010). As shown in **Fig. 6**, the air temperature at the surface is directly affected by the SSTs and there is a strong cold deviation of more than 5 ℃ in the WRF-ROMS in the Sea





of Okhotsk and Kuril Islands during the extended period. The errors of air temperature at the surface in the Sea of Okhotsk and Kuril Islands of WRF-SOM are within 3 °C in the extended period, which is much closer to the CFSv2 reanalysis compared with WRF-ROMS.

Since the main deviation between WRF-ROMS and WRF-SOM mainly comes from the sea surface, in order to explore the influence of SST on the whole atmosphere, we study the variation of RMSEs of air temperature and geopotential height (GPH) with different heights to characterize the stability of the subtropical high and the upper atmosphere (Lu and Lin, 2009; Zhou and Yu, 2009). The RMSEs of air temperature increase with height, and the differences between the two models are the biggest at the surface (**Fig. 7a**). The deviation of air temperature gradually disappears when reaching the height of 300

hPa (**Fig. 6b**). The RMSEs of the GPH also increase with the height (**Fig. 8a**). However, the differences between the two models are opposite to the temperature and increases with the height (**Fig. 8b**). Compared with the WRF-ROMS, the WRF-SOM performs better in the forecast of the GPH field in the high, middle, and low atmosphere, as shown in **Fig. 8**. The difference between the RMSEs of GPH in the two models are increasing from the lower level to the upper level, which means that the deviation between the WRF-SOM and the WRF-ROMS is generated from the surface and propagates to the

upper layer. Therefore, combined with the results of air temperature and GPH, the response of variables with different physical properties to SST will also appear in different states. In terms of the extended-range prediction, WRF-SOM has obvious advantages in the areas around the Kuril Islands and the Sea of Okhotsk, where WRF-ROMS has large deviation in SSTs.

## 3.3 The prediction of tropical cyclones in extended-range scales

Typhoon is an important extreme weather phenomenon in the extended-range forecast, and the typhoon in the Western Pacific has a profound impact on coastal countries (Webster et al., 2014). The typhoon processes are deeply affected by the air-sea interaction, which can range from days to weeks. Therefore, typhoons are selected as an example of the extreme weather to discuss the atmospheric predictability in the extended period. Following previous studies (Webster et al., 2014), we use track and intensity as the key prediction parameters. As the typhoon simulation from August to October 2020 (**Fig.**

**9a-9j**), the WRF-SOM can also obtain slightly better prediction paths than the WRF-ROMS after abandoning the ocean dynamics framework during the typhoon season. The results of typhoon tracking in the WRF-SOM are better than those in the WRF-ROMS within 72 hours during the processes of typhoons, as shown in **Table 1**. The simulation of typhoon tracks is mainly dominated by steering flow in the atmosphere model, and the improvement of SST can only slightly optimize the path (Anthes, 1982; Hollland, 1983) such that the forecast results are similar in WRF-SOM and WRF-ROMS. **Figure 10a-**

**10j** show the performances of MWS of 11 typhoons from August to October in 2020, both in WRF-SOM and WRF-ROMS. Both systems are unable to achieve accurate simulation for super typhoons exceeding 40 m/s. However, BAVI has better MWS performances in WRF-SOM than in WRF-ROMS, as shown in **Fig. 10e**. We find that in the process of model simulation, the typhoon MWS is positively correlated with the SST, which can be caused by the surface heat flux and the surface water vapor. Among the eleven typhoons including three super ones, there is little difference in typhoon MWS



between the WRF-SOM and the WRF-ROMS, which means that both the 3-D ocean dynamical ocean model and the SOM have defects on the simulation of high-intensity typhoons. As for typhoon simulation, WRF-SOM can obtain comparable prediction results with WRF-ROMS.

## 4 Summary and discussion

In this study, to improve the numerical model predictability of monthly extended-range scales, we use the simplified SOM to
restrict the SST bias. It is because the 3-D dynamical ocean model inevitably introduces unnecessary biases from the dynamics and seabed topography. Therefore, the experiments are implemented with the WRF-ROMS and WRF-SOM to investigate the SST deviation in the extended-range period and the associated atmosphere responses. We systematically evaluate the SST prediction effect of the WRF-SOM and the WRF-ROMS against the HYCOM reanalysis. As for SST prediction, whether in space or time, the performance of the WRF-SOM is better than that of the WRF-ROMS, especially in
the Okhotsk Sea and the area north of 15°N. WRF-SOM can effectively avoid the deviation in the deep layer from 3-D dynamical ocean models. Furthermore, the reduction of SST biases in the WRF-SOM has a significant impact on the atmosphere at the surface, which not only affects the air temperature but also indirectly changes the GPH field in the middle and upper layer of the atmosphere. The WRF-SOM can obtain the compatible typhoon path and maximum wind speed predictions with WRF-ROMS and reduce the consumption of computing resources by roughly 50%.

It is shown by our experiments that the subsurface modeling errors in the 3-D dynamical ocean model could propagate to the surface with the forecast lead time and make a large deviation in SST. To improve the predictability in the extended period, it is of vital importance to constrain the deviation of SST. Based on the good persistence of SST, it is verified that using the SOM instead of the 3-D dynamical ocean model can have a better prediction skills and save a lot of computing resources. For the extreme weather event such as typhoons, the predictions of WRF-SOM are in good agreement with WRF-ROMS.
However, the WRF-SOM also has its own limitations. The overall simulation of SST in WRF-SOM is relatively stable. Due to the abandonment of the dynamical framework, the WRF-SOM may not be able to obtain ideal prediction results in some areas dominated by local dynamic processes (e.g., surface currents, vortex, and turbulence).

Considering the SST characteristics in the extended-range predictions and the limitation of available computing resources, our method provides a new idea for exploring the predictability in the extended period. At present, our prediction
experiments cover summer, autumn, and the first half of winter, which leads to the lack of representation of other seasons. Moreover, we do not pay too much attention to the underlying surface temperature before typhoon generation in this study, but it is an important driving factor for typhoon generation predictions. In future, it is useful to expand the number of prediction examples to cover a longer period such as one year, extend the forecast time of each case, and improve the model horizontal resolution, and further get insights on the WRF-SOM in the predictability of typhoon genesis. Finally, due to the
joint impact of the initial conditions and the external forcing on the extended-range predictability of the atmosphere, we need to add the control experiments to quantitatively evaluate the effect of nonlinear errors growth in the atmosphere and external



forcing differences from the ocean on the extended-range predictions.

**Code and data availability**

Codes, data, and scripts used to run the models and produce the figures in this work are available
on the Zenodo site (https://doi.org/10.5281/zenodo.6630331, Wang et al., 2022) or by sending a written request to the
corresponding author (Shaoqing Zhang, szhang@ouc.edu.cn).

**Author contributions**

Zhenming Wang is responsible for all plots, initial analysis, and some writing; Shaoqing Zhang proposes the idea; Shaoqing
Zhang leads the project, organizes and refines the paper; Yishuai Jin, and Yinglai Jia provide significant discussions and
inputs for the whole research; all other co-authors make contributions by wording discussions, comments, and reading proof.

**Competing interests**

The authors declare that they have no conflict of interest.

**Acknowledgments**

The research is supported by the National Natural Science Foundation of China (Grant No. 41830964) and Shandong
Province's "Taishan" Scientist Program (ts201712017) and Qingdao "Creative and Initiative" frontier Scientist Program (19-
3-2-7-zhc). All numerical experiments are performed on the platforms at Qingdao Pilot National Laboratory for Marine
Science and Technology.

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











Figure 1: Schematic illustration of a Slab Ocean Model (SOM) coupled to the Weather Research and Forecasting model (WRF). $\tau_x$ $\tau_y$ are respectively the zonal and meridional component of wind stress at the surface, $h_u$ ($h_v$) is the $\tau_x$-driven ($\tau_y$-driven) momentum in the ocean mixed layer, $f$ is Coriolis force, $C_0$ is the specific heat capacity of the ocean water, and $h_{mix}$ is the mixed layer depth.



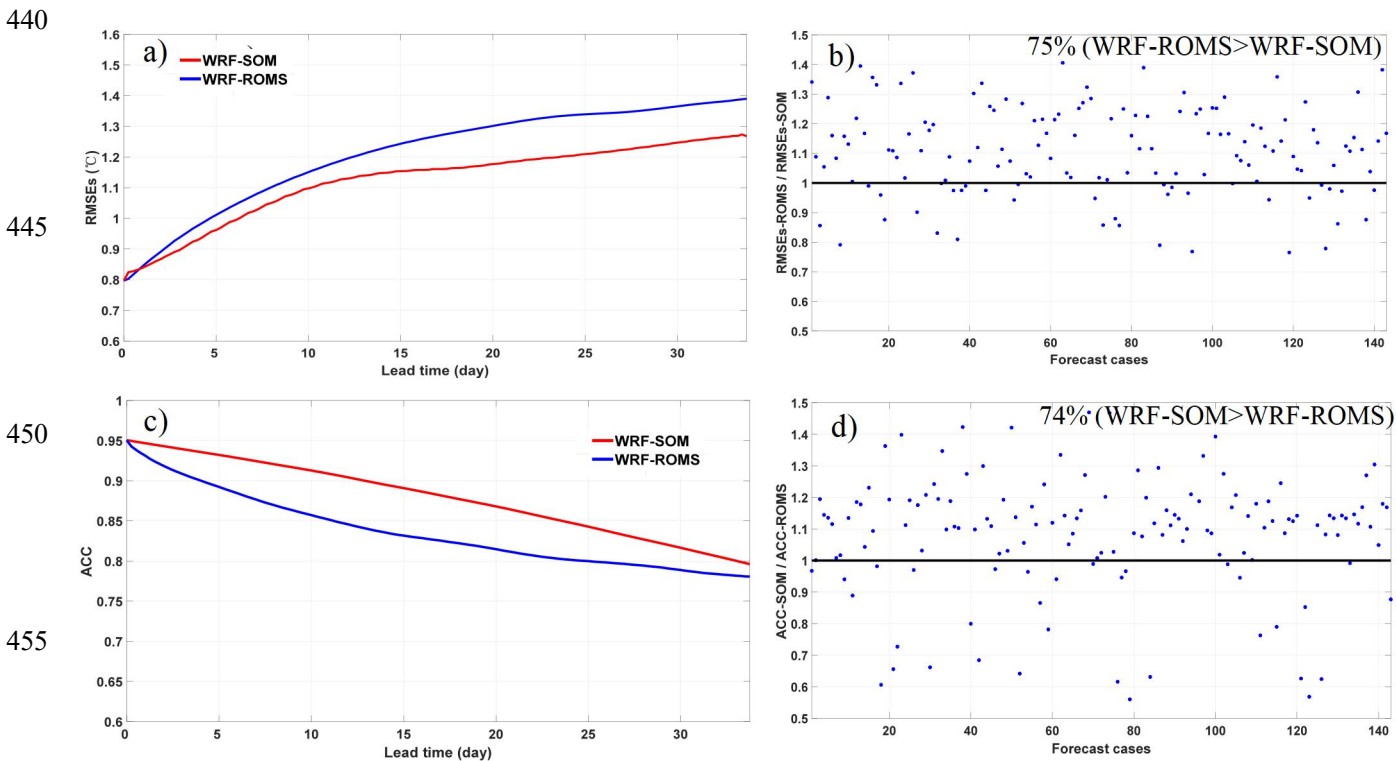

**Figure 2: Time series of *a*) averaged root mean square errors (RMSEs), and *c*) anomaly correlation coefficients (ACCs) of simulated sea surface temperatures (SSTs) against Hybrid Coordinate Ocean Model (HYCOM) reanalysis of total 142 forecast cases from July 19ᵗʰ to December 31ˢᵗ, 2020. The comparison of the *b*) RMSEs, and *d*) ACC between WRF-SOM and WRF-ROMS for each case.**

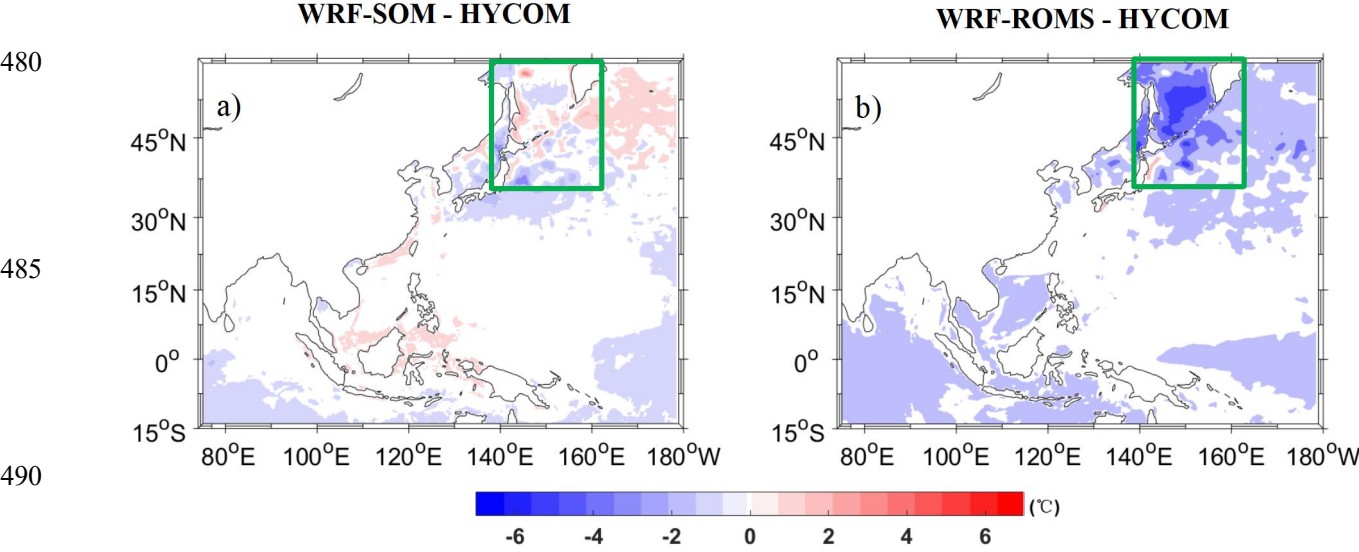

**Figure 3: The spatial distributions of the SST errors of *a*) WRF-SOM, and *b*) WRF-ROMS against the HYCOM reanalysis of total 142 forecast cases from July 19th to December 31st, 2020 averaged in the 34-day forecast period. The region in green rectangle (35° N-58° N, 140° E-160° E) is the cold deviation area in WRF-ROMS.**



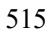

**Figure 4: The spatial distributions of ACC of forecasted SSTs in the WRF-SOM (left column, panels *adg*), WRF-ROMS coupled models (middle column, panels *beh*) and their comparisons of each grid (right column, panels *cfi*) in the model domain (green dots) including cold deviation area (red dots) against HYCOM reanalysis, averaged in the first 10 days (upper panels *abc*), days 11-20 (middle panels *def*), days 21-30 (bottom panels *ghi*) forecasts of total 142 forecast cases from July 19th to December 31st, 2020.**



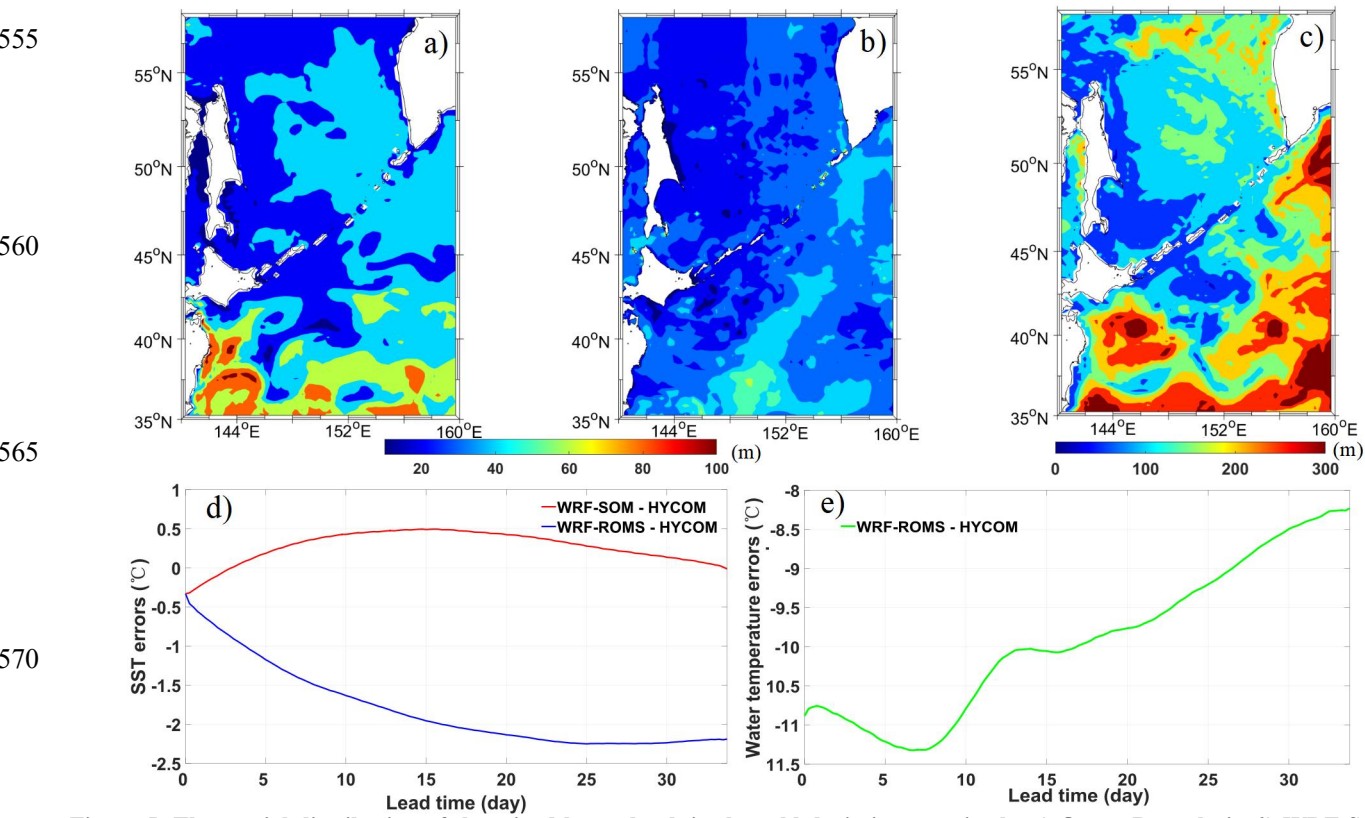

**Figure 5: The spatial distribution of the mixed layer depth in the cold deviation area in the *a*) Ocean Reanalysis, *b*) WRF-SOM, and *c*) WRF-ROMS of total 142 forecast cases from July 19th to December 31st, 2020 averaged in the 34-day forecast period. The time-series of averaged water temperature errors at the *d*) surface in WRF-SOM (red) and WRF-ROMS (blue), and the *e*) subsurface in WRF-ROMS (green) against HYCOM reanalysis in the cold deviation area.**



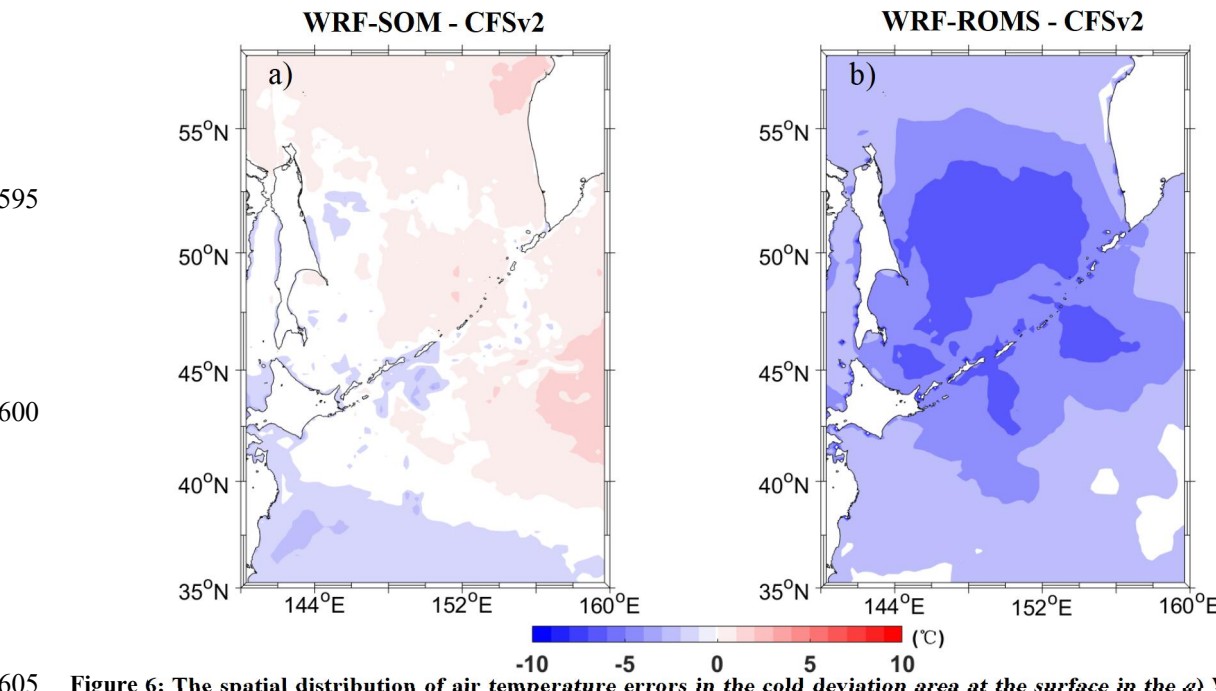

Figure 6: The spatial distribution of air temperature errors in the cold deviation area at the surface in the a) WRF-SOM, and b) WRF-ROMS against Climate Forecast System versions 2 (CFSv2) reanalysis of total 142 forecast cases from July 19[th] to December 31[st], 2020 averaged in the 34-day forecast period.





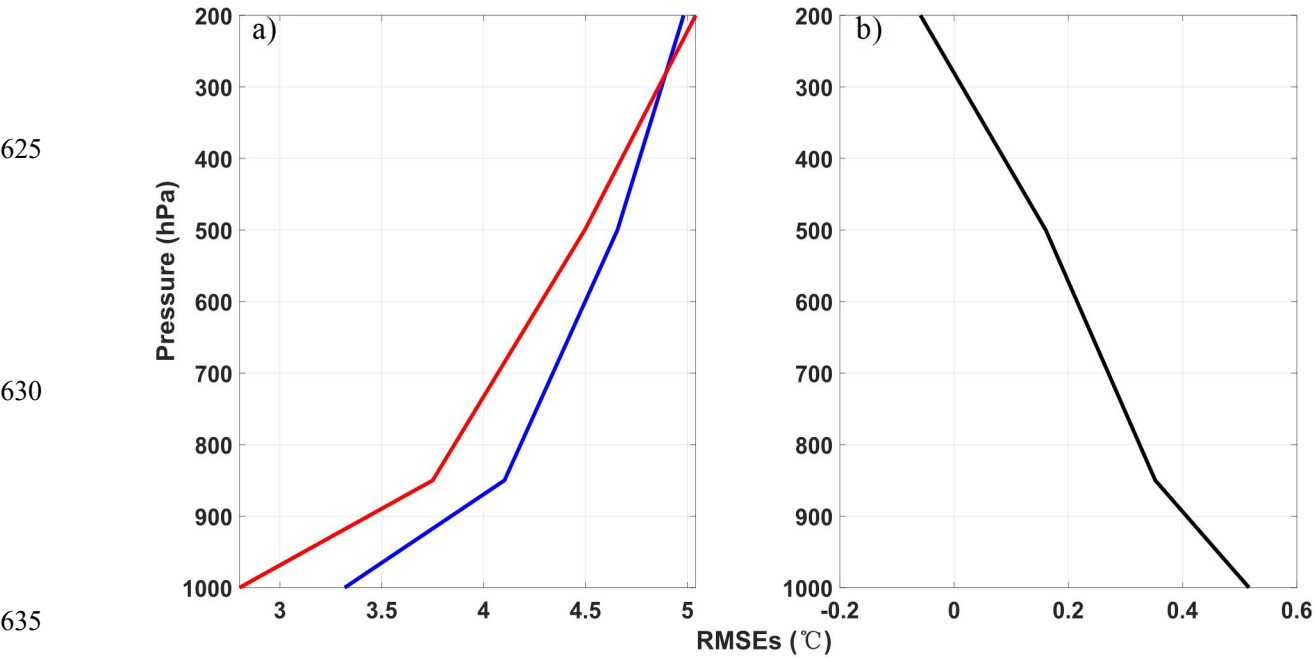

**Figure 7: The *a*) variations of RMSE of air temperature with air pressure between the WRF-SOM and WRF-ROMS against the CFSv2 reanalysis averaged in the 34-day of total 142 forecast cases from July 19th to December 31st, 2020, and the *b*) variation of the difference of RMSEs in two models with the air pressure.**

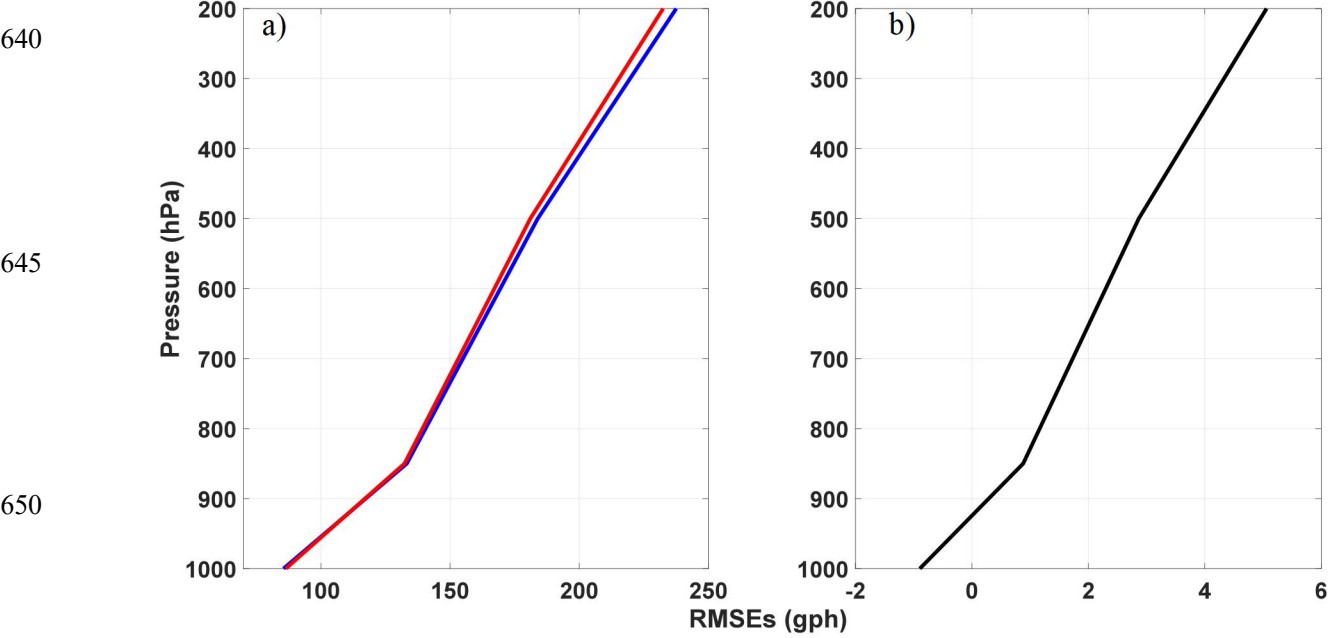

**Figure 8: The *a*) variations of RMSEs of the geopotential height (GPH) with air pressure in the WRF-SOM (red) and WRF-ROMS (blue) against CFSv2 reanalysis of total 142 forecast cases from July 19th to December 31st, 2020, and the *b*) variation of the difference of RMSEs in two models with the air pressure.**



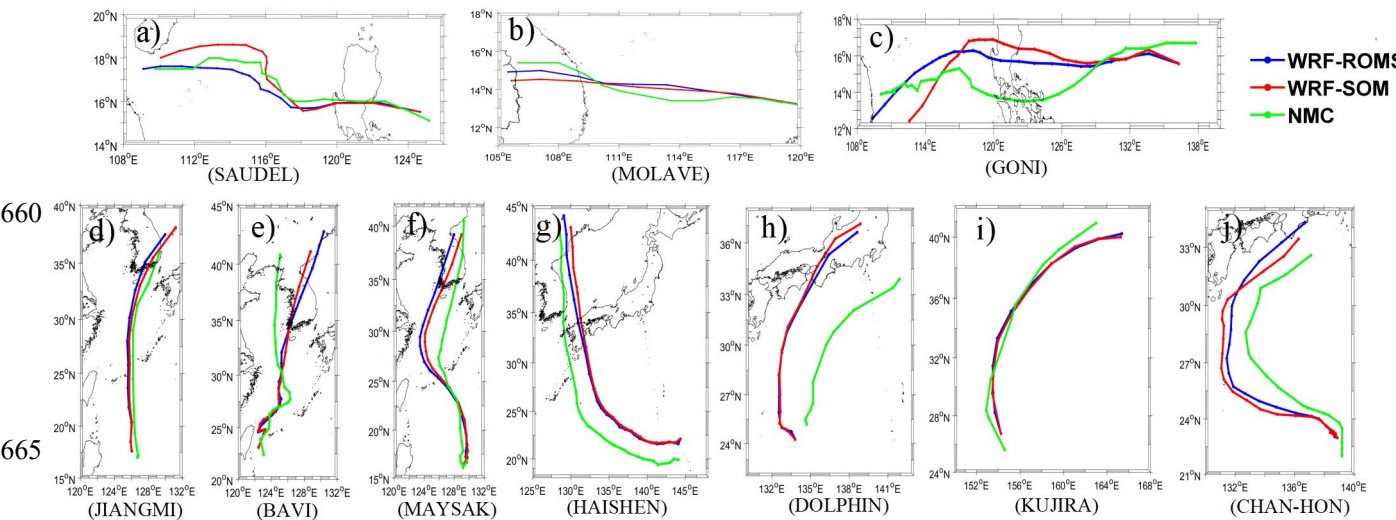

**Figure 9: The typhoon tracks simulated in the WRF-SOM (red) and WRF-ROMS (blue) compared with National Meteorological Center (NMC) (green) during typhoon season (NMC data, http://typhoon.nmc.cn/web.html).**

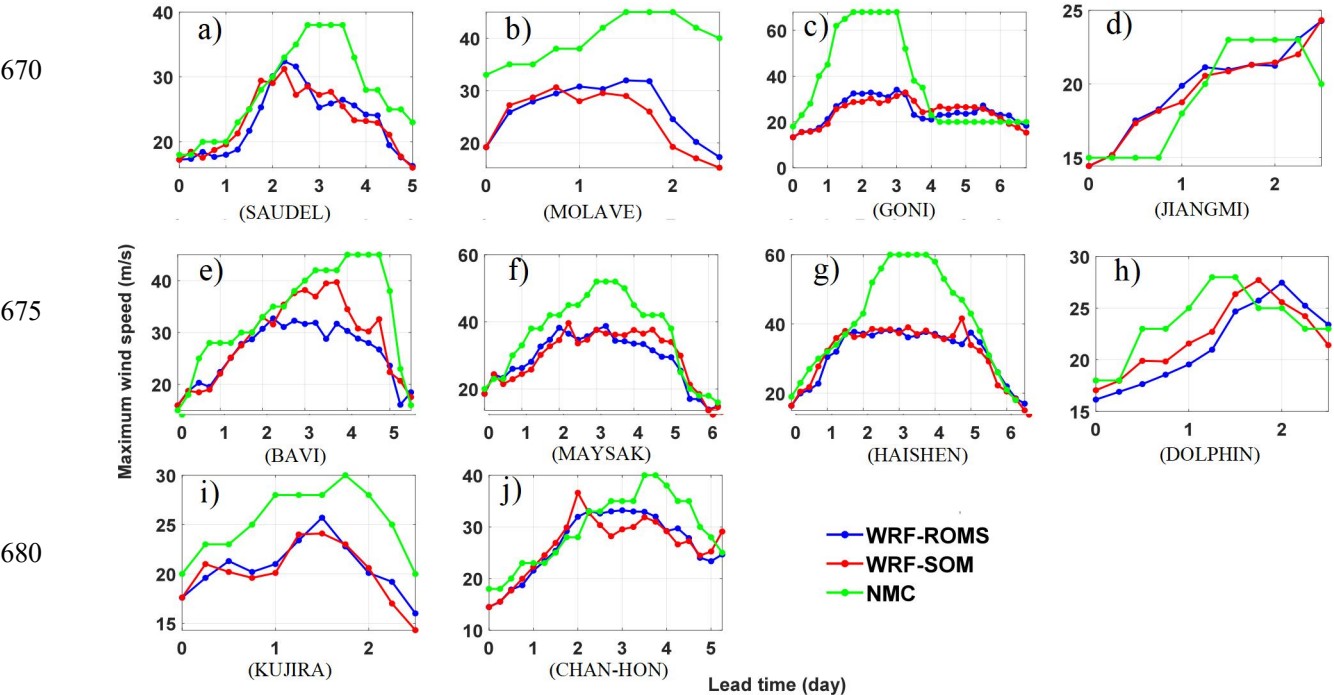

**Figure 10: The maximum wind speed (MWS) of typhoon simulated in the WRF-SOM (red) and WRF-ROMS (blue) compared with National Meteorological Center (NMC) (green) during typhoon season (NMC data, http://typhoon.nmc.cn/web.html).**



**Table 1: Typhoon track errors in different simulation periods compared with observations from NMC**

| Lead time (hour) | Model | The distance of typhoon center against observations (km) |
|---|---|---|
| 24 | WRF-SOM | 171 |
|  | WRF-ROMs | 187 |
| 48 | WRF-SOM | 188 |
|  | WRF-ROMs | 204 |
| 72 | WRF-SOM | 224 |
|  | WRF-ROMs | 247 |

690