# Peer review of "Monthly-Scale Extended Predictions Using the Atmospheric Model Coupled with a Slab-Ocean"

_Geoscientific Model Development, 2022_

## Author Response (AR1)

A point-by-point response to the review 1:

This study compares the month-scale predication with WRF coupling with two different modes: Slab-ocean model (SOM) or ROMS. It shows that the SOM performs better than the ROMS by avoiding the SST bias in ROMS. The result are interesting and the topic fits GMD well. The experiments are well designed and manuscript is well written. I suggest major revision for with my comments listed below:

**RE:** Thanks for the reviewer's thorough examination of our manuscript (MS) and comments. All of coauthors agree that the comments are very constructive to improve the presentation of the MS, and all the major comments and other points have been fully addressed in the revision. Specifically, in the revision, we have added: 1) more detailed descriptions of the WCDA, 2) SST prediction skills over different seasons in Figure 2e, 3) more detailed descriptions of the possible error sources, etc.

The point-by-point replies are followed.

Your model domain covers north Pacific high latitude. Some of the area should be covered by sea ice, especially in winter. How do you deal with the sea ice in your SOM? The sea-ice region overlap with the region with large cold SST error in (Figure 3). What is the forecast sea ice error in ROMS?

**RE:** Yes we agree that the sea-ice region overlap with the region with large cold SST errors and we do believe that the inaccuracy of sea ice simulation would aggravate the SST errors. Firstly, SOM does not calculate sea ice separately and it reads sea ice data from CFS data every six hours. For ROMS, the regional prediction system does not include a separate sea-ice module. We have added the detailed descriptions of dealing with the sea ice in the revision. Please see lines 218-222. Figure 5e shows that the errors of SST errors are mainly from the subsurface in this area at the start of the simulation. This paper mainly aims to propose SOM to avoid the complex errors from

dynamic processes and inappropriate model configuration. Thanks for your good suggestions and we will evaluate the influence of sea ice on the SST predictions in the extended-range predictions in the follow-up work.

Section 2.3 Can you provide some details about your WCDA in terms of DA method and assimilated observations? The details of generating initial conditions of SOM are missing, for example, how do you derive the mixed layer depth.

**RE:** Thanks for your good suggestion. We have added the details of the WCDA in the revision. Please see lines 151-155. The observations assimilated in the forecast system including: 1) GTS observation packages for the atmosphere (air pressure, wind, geopotential height and temperature) are from China Meteorological Administration. 2) The observations for the ocean (ocean temperature and salinity) are from AVHRR, OSTIA, ARGO, and AVISO, etc.

In SOM, we calculate the depth of the mixing layer by the joint action of surface wind stress and heat budget of the ocean mixing layer. Please see Equation 1-4.

It will be good to show the prediction improvement over different season.

**RE:** Thanks for your suggestions. We have added the prediction improvement over different seasons in Figure 2e and relevant analysis. Please see lines 207-211.

Ln 148 "for each example" to "for each experiment"

**RE:** Thanks for your suggestions. We have changed the statement from "for each example" to "for each experiment". Please see line 164.

A point-by-point response to the review 2:

In this very interesting study, the author has compared the performance of extended-range forecasts in WRF-ROMS against a similar system but replaced the 3D ocean with a slab ocean model. Comparing SST and TC forecasts between these two systems suggests that a coupled forecasting system with a rather simple slab ocean model can do a decent, or sometimes even better job than that with a full 3D dynamic ocean. These results are scientifically interesting. However, the presentation of this work needs some improvements. Therefore, I suggest acceptance after major revision. Please see detailed comments below.

**RE:** Thanks for the reviewer's thorough examination of our manuscript (MS) and comments. All coauthors agree that the comments are very constructive for us to improve the presentation of the MS, and all the comments have been fully addressed in the revision. We have weakened the strong statement of SOM, and propose SOM as an effective method for extended-period prediction. Specifically, in the revision, we have added: 1) more detailed descriptions of WCDA, 2) more detailed descriptions of heat budget calculation of SOM, 3) the reason for using reanalysis (HYCOM) as the validation, etc.

The point-by-point replies are followed.

In this study, the authors argue that coupled (extended-range) forecasts with a slab ocean model performed better than a similar system but using a 3D dynamical ocean model. This argument may be true for the setup used by this study, but need to be proven for other system, and shouldn't be presented as a general statement for all coupled forecasting systems. There is no doubt that using a slab ocean has some benefits w.r.t 3D dynamic ocean modelling (e.g. reduced computing cost, or sometimes, even reduce SST bias if the 3D ocean model is not well configured). However, it is generally believed that you do need a 3D ocean model and a fully

coupled system to achieve better forecasts. A good example is the tropical cyclone forecasts studies carried out by Mogensen et al (2018). TCs interact with the SST in three ways: the heat transport to the atmosphere e.g., the vertical mixing with deeper water e.g., and the upwelling generated by Ekman pumping e.g.. While the first (resp. second) process could be easily represented by using a slab ocean model (with the help from a 1D mixed-layer model), a three-dimensional model is nonetheless required to properly represent all three processes.

Therefore, I suggest to play down this strong statement (that a slab-ocean model is overall superior than a 3D ocean model in the sense of making better extended-range forecasts) in this manuscript. Instead, this work can be presented as a study to prove that even a simplified coupled forecasting system with a slab ocean model can work effectively, or in some cases, better than a fully coupled forecasting system using a 3D ocean model.

**RE:** Thanks for your good suggestions. We agree that more prediction systems are needed to verify this conclusion, which will be put into our follow-up work. Therefore, we have played down the strong statement in the revision. Please see lines 290-292. The interaction between the typhoon and the ocean really needs the support of a 3-D dynamical ocean model, but focusing on the prediction of typhoon generation, a stable SST variation in the SOM may be an advantage. Please see lines 307-309. There is no doubt that the 3-D dynamical ocean model is necessary for the development of forecast skills, and the WRF-SOM is verified as an effective way to relieve the rapid error growth in some regions. We have added the relevant analysis. Please see lines 299-301.

Other general comments

Design of experiments: it is suggested to extend your experiment period (6 months is a bit short) and switch to a slightly different forecasting interval (e.g. 1-wk instead of everyday) when launching forecasts. I understand that it requires significantly amount of work so I leave it to the authors to decide whether or not to carry out this work. But

it is worth to point out that results in this study are subject to a relative short testing period that only occupies the second half of the year (no boreal spring or first half of summer season).

**RE:** Thanks for your suggestion. The forecast experiments that last for half a year are really large workload. At present, we have added the seasonal variation of SST error in Figure 2e and relevant analysis. Please see lines 207-211. We will certainly implement your suggestion in the follow-up work to cover the prediction that lasts for one year or even longer.

Methodology: Section 2.3 need more details, particularly about your DA method (WCDA) used in this study and your experiment setup. E.g. do you use the same WCDA for both WRF-ROMS and WRF-SOM systems? How do you ensure that initial and boundary conditions in both systems are the same?

**RE:** Thanks for the good suggestion. We have added the details of the WCDA in the revision. Please see lines 151-155. We have not adopted WCDA in WRF-SOM. In order to ensure the consistency of initial conditions in the WRF-SOM and WRF-ROMS, the initial conditions are obtained from the global forecast system, while the boundary conditions are both interpolated from NCEP-CFS.

Verification: If possible, one should always try to verify forecasts against observation. In the sense of SST forecasts, there are plenty of observation based SST product available, e.g. L4 products like GHRSST, OIv2 SST, ESA CCI SST. Verification against these products should be encourage instead of against HYCOM reanalysis. If the authors chose to use other reanalysis product (e.g. HYCOM) as verification data set, then at least the reason behind this choice should be added.

**RE:** Thank you for your suggestion. Firstly, it is true that there are many observations of SST and there is no doubt observations are much more convincing than the reanalysis. But it is difficult for us to obtain ocean observations with appropriate time interval and high horizontal resolution. Compared with that the advantages of HYCOM are very suitable to us. For instance, the horizontal resolution can reach

0.08 ° and its time interval matches our prediction system for six hours. Secondly, HYCOM has been systematically verified by many scholars (EJ Metzger et al., 2008 and Wallcraft et al., 2007). In addition, the reanalysis output is more stable than the observation, and there is almost no missing for reanalysis. Based on the above reasons, we choose HYCOM as the standard to evaluate the forecast system.We have added the reason behind this choice and we would pay more attention to observations as verification in the further study. Please see lines 167-170. We have added the responding references in the revised manuscript. Please see lines 391-397.

Specific Comments

L41: In the extended period prediction, SST is the most important …

This is rather strong statement. SST is no double an important part of air-sea interaction, but other element, e.g. sea-ice condition, is also crucial for accurate extended-range forecast.

**RE:** Thanks for your advice. At the beginning, we just want to emphasize the importance of SST and we have corrected the statement of the sentence. Please see lines 41-42.

L52-56: When discussion potential issues related with coupling to 3D ocean model, it is suggested to point out that any issue (as suggested in Wu et al., 1997 and Ren and Qian 2010) can be specifically related with the ocean model used in their studies. These issues are normally specific to their system configurations – e.g. model resolution, boundary conditions, model parameterisation, numerical stepping, etc), and can be improved by, e.g. using a better bathymetry input if the model bias is directly related with a sub-optimal bathymetry file. L56-58: Again I believe that it is not a general issue in dynamic 3D ocean modelling, but rather an issue specific to the set up in that study carried out by Hu et al., 2017.

**RE:** Thanks for your suggestions. we agree that the issues in 3-D ocean model is specifically related with the system configurations. The 3-D model does have its

inherent error source caused by the difference scheme and dynamic processes (Wu et al., 1997) and our purpose for citing these two articles (Hu et al., 2017 and Ren and Qian 2010) is to point out the difficulty of the error control in 3-D dynamical ocean models. We have modified the sentences. Please see lines 62-63.

L61-63: I suggest to restrain from saying that SST is the "most" important factor provided by the ocean.

**RE:** Thanks for your advice. We have corrected the statement of the sentence. Please see lines 70-71.

L91-94: Please rephase this sentence to clarify what you mean.

**RE:** We have modified the sentence to clarify the the basic situation of the forecast system. Please see lines 100-105.

L100: SOM was referring to Slab Ocean Model, here was referred to as "a simple model"

**RE:** Thank you for pointing out the problems. This is indeed an expression easy to cause ambiguity. We corrected the sentences. Please see line 111.

Eq1: how do you compute Q_ocn in SOM? Does SOM treat subsurface as a bottom boundary condition from CFSv2 reanalysis ?

**RE:** In SOM, there is a fixed temperature lapse rate below the mixing layer, and Qocn is calculated through heat exchange with subsurface layer. SOM is a simple model with only one layer and does not contain the information of subsurface from CFSv2. We have added the Q_ocn description in the revision. Please see lines 117-118.

L130: typos with two "from"

**RE:** Thanks for pointing out the problems. We have corrected the sentence. Please see line 141.

L136: Do you mean you use CFSv2 forecasts as boundary condition for WRF-ROMS spun-up? Why not use CFSv2 reanalysis?

**RE:** Thanks for your good question. The WRF-ROMS is an operational forecast system. Due to the limit of reanalysis data, boundary conditions can only be obtained from forecasts and the initial conditions and boundary conditions extracted from the CFSv2 forecast also containing the observation information.

L138-140: Could you elaborate a bit on the WCDA approach taken in your experiment? What's obs are assimilated and how do you set up your WCDA framework? This is a rather important component for any forecasting system.

**RE:** Thanks for the good suggestion. We have added the details of the WCDA and the basic process of data assimilation of the atmosphere and the ocean in the revised manuscript. We have modified the inappropriate expression. Please see lines 151-155.

L158: .. is evaluated by the root mean square error (RMSE) "and" the anomaly correlation coefficient (ACC), …

**RE:** Yes. We have modified the sentence, please see line 175.

Eq5 and 6, f_ij can be obs or analysis data, but not the truth value (which we never know).

**RE:** Thanks for your advice. We have modified the description of the variable in Eqs 5 and 6. Please see line 183.

Section 3.1 first paragraph can be moved to Introduction

**RE:** Thanks for your advice. We have moved first paragraph in Section 3.1 to Introduction. Please see lines 52-69.

L193-194: in Figure 3a and 3b, cold bias in WRF-ROMS in the green box can be an issue related with, e.g. inappropriate boundary conditions and/or lack of sea-ice model in your system, particularly during the winter season in the north hemisphere at this

latitude. Reason that SOM is doing better may simply because it lacks 3D ocean advection to propagate this cold bias from boundary to other regions. Or, may be the upper ocean mixing is over-estimated in the WRF-ROMS experiment, leading to systematically cold bias almost everywhere. These possibilities should be checked/explained.

**RE:** Thanks for your good comments and suggestions. We all agree that the lack of sea-ice module in the WRF-ROMS will affect the heat budget at the sea surface in winter. The WRF-SOM has read the sea ice data from CFS in the SOM scheme, which by some degree relaxes the issue. We have shown that the error of WRF-ROMS comes from the subsurface at the start of the simulation in Figure 5d and 5e. The WRF-SOM only includes the basic air-sea interaction at the ocean surface, and therefore the cold deviation is much smaller. We have added the relevant analysis in the revision. Please see lines 218-222. Based on the comments, we have added more relevant analysis of error sources for the WRF-ROMS and WRF-SOM. Please see lines 238-242.

L205: suggest to replace predictability by "mean biases"

**RE:** Thanks for your good suggestions. We have modified the sentence. Please see line 225.

L206-215: I understood that definition of mixed layer depth is quite different between WRF-SOM (wind stress and surface heat determined) WRF-ROMS (0.2C from SST) and HYCOM reanalysis (not sure, need add this information). This makes it difficult to interpolate results at Fig. 5, as various MLD definitions can diff as much as 50-100m.

**RE:** Thanks for your good suggestions. The depth of mixed layer in HYCOM is also calculated based on 3-D ocean temperature (0.2C from SST). We have added the information. Please see lines 235-236.

L210: What do you mean by reanalysis data from ECMWF?

**RE:** Thanks for pointing out the problem. We have corrected the mistake of words. Please see line 237.

L211: Please add information to define the so called "subsurface" in Fig 5-e (spatially averaged temperature below mixed layer depth in WRF-ROMS?). SST in WRF-ROMS is systematically colder than WRF-SOM is a results directly related with their MLD differences. Atmospheric heat fluxes warm the surface ocean in winter season (the experiment period), a relatively shallower MLD in WRF-SOM means more heat resides in the mixed layer, leading to higher SST. Lack of vertical convection and Ekman pumping effect in the slab-ocean model can enhance this surface heat residence effect even more in WRF-SOM forecasts.

**RE:** Thanks for your good suggestions. Subsurface in this paper refers to the area between 300m-400m depth. We show the temperature error at the subsurface to determine the source of cold deviation. We have added the description of subsurface. Please see the caption of Figure 5. We agree that lack of vertical convection and Ekman pumping effect in the SOM can enhance this surface heat residence effect and we think that if it is not for extreme weather, these two factors do not play a leading role in the prediction of SST. We have added the relevant analysis in the revision. Please see lines 238-240.

L213-215: … the data assimilation can accelerate the heat loss and intensify the cooling in this area… Could you please elaborate a bit here? In general, data assimilation adds constraint to the ocean state variables (temperature/salinity, e.g.) towards observations. Performance of analysis (which is used to initiate your forecasts as I understood) is subject to your model biases, boundary conditions, efficiency and effectiveness of DA method, as well as the quality and quantity of your input observation streams. It is unusual to claim that DA has degraded your analysis performance w.r.t your free run (without DA), which basically means that your DA

system is not working as intended.

**RE:** We agree that he sentence "… the data assimilation can accelerate the heat loss and intensify the cooling in this area…" is very inappropriate. We have removed inappropriate statement and added the possible causes of errors. Due to the complexity of the heat budget in the Okhotsk Sea, the data assimilation system that could work reasonably in other regions would be somewhat inappropriate here, which may require further confirmation through detailed assessment of the heat budget in this area in the follow-up work. We have added the relevant descriptions. Please see lines 240-242.

The statement "by eliminating the influence of initial conditions and ocean heat transport …" sounds like you are suggesting that a forecasting system initialized from a free run (without DA) works better than that initialized from a analysis (with DA), which is not consistent with the modern NWP system.

**RE:** Thanks for your good suggestions. The original intention of the sentence "by eliminating the influence of initial conditions and ocean heat transport …" is to eliminate the influence of the initial field by forecasting from the same initial conditions. We are sorry to have caused you misunderstanding. We have deleted the ambiguous words. Please see line 238-242.

---

## Author Response (AR2)

A point-by-point response to the Topical Editor:

Comments to the author:

Although two reviewers have suggested minor revisions, I strongly suggest that the authors should briefly explain the larger SST RMSE of WRF-SOM forecast initialized in July.

RE: Thanks for your good comments and suggestions. We have explained the reason for the large SST errors in July. Please see lines 145-147 and 203-206.

Also, please discuss the associated impact on the typhoon forecasts in July. Currently, only cases from August to October were chosen.

RE: Thanks for your advice. Since there is no typhoon observed in the north-west Pacific in July 2020, we can not do the relevant analysis. We do believe these SST errors in the region could have effect on the typhoon activities and we will expand the experiments to explore its impact in the further work. Please see lines 280-283.

Some minor comments:

Line 152: please remove "local"

RE: Thanks for your good suggestions. We have removed the word. Please see line 155.

Line 240: please cite proper references to indicate the potential sources for the cold biases.

RE: Thanks for your suggestions. We have added the proper references in the paper. Please see lines 237-238 and 414-420.

A point-by-point response to the review 1:

Most of my previous comments have been addressed in the revised version. I only have the following minor comment in this round of review.

The effect of sea ice is important on the SST predictions during the extended period which did not include in this study. I suggest more discussion or speculation on this issue in the paper. I also encourage authors deeply study the impact of sea-ice parameterization in the SOM in future work.

RE:Thank you for your suggestions. We have added the discussion on sea-ice and emphasized the importance of sea-ice in the follow-up work. Please see lines 208-210.

A point-by-point response to the review 2:

Only a few minor comments

L147: The boundary condition of the forecast is interpolated from the CFSv2 forecast data set.

So both system uses the same CFSv2 forecasts as boundary conditions?

RE:Yes. Both systems use the same boundary conditions from CFSv2 forecast.

L289: 3-D dynamical ocean model inevitably introduces "unnecessary" biases …

Suggest to rephase. I would not say unnecessary biases.

RE:Thank you for your suggestions. We have rephrased this sentence. Please see line 286.

L293: … can manage with the "easier" errors than 3-D dynamic …

Not sure what easier mean, please rephase

RE:Thank you for your suggestions. We have rephrased this sentence. Please see line 290.

L306: .. by local dynamics processes (…

Suggest to add "Ekman pumping"

RE: Thank you for your suggestions. We have add "Ekman pumping" in the dynamic processes. Please see line 304.